# Analysis of the alpha activity envelope in electroencephalography in relation to the ratio of excitatory to inhibitory neural activity

**Misako Sano[1], Yuko Nishiura[1], Izumi Morikawa[1,2], Aiko Hoshino[1], Jun-ichi Uemura[1], Katsuyuki Iwatsuki[3], Hitoshi Hirata[3], Minoru Hoshiyama[1]** *

**1** Department of Preventive Rehabilitation Sciences, School of Health Sciences, Nagoya University, Nagoya, Japan, **2** Music Division, Nagoya University of the Arts, Kitanagoya, Japan, **3** Department of Hand Surgery, Graduate School of Medicine, Nagoya University, Nagoya, Japan

* hosiyama@met.nagoya-u.ac.jp

**Data Availability Statement:** All data used are available from the open datasets, https://openneuro.org/datasets/ds004504/versions/1.0.6,

## Abstract

Alpha waves, one of the major components of resting and awake cortical activity in human electroencephalography (EEG), are known to show waxing and waning, but this phenomenon has rarely been analyzed. In the present study, we analyzed this phenomenon from the viewpoint of excitation and inhibition. The alpha wave envelope was subjected to secondary differentiation. This gave the positive (acceleration positive, Ap) and negative (acceleration negative, An) values of acceleration and their ratio (Ap-An ratio) at each sampling point of the envelope signals for 60 seconds. This analysis was performed on 36 participants with Alzheimer's disease (AD), 23 with frontotemporal dementia (FTD) and 29 age-matched healthy participants (NC) whose data were provided as open datasets. The mean values of the Ap-An ratio for 60 seconds at each EEG electrode were compared between the NC and AD/FTD groups. The AD (1.41 ±0.01 (SD)) and FTD (1.40 ±0.02) groups showed a larger Ap-An ratio than the NC group (1.38 ±0.02, p<0.05). A significant correlation between the envelope amplitude of alpha activity and the Ap-An ratio was observed at most electrodes in the NC group (Pearson's correlation coefficient, r = -0.92 ±0.15, mean for all electrodes), whereas the correlation was disrupted in AD (-0.09 ±0.21, p<0.05) and disrupted in the frontal region in the FTD group. The present method analyzed the envelope of alpha waves from a new perspective, that of excitation and inhibition, and it could detect properties of the EEG, Ap-An ratio, that have not been revealed by existing methods. The present study proposed a new method to analyze the alpha activity envelope in electroencephalography, which could be related to excitatory and inhibitory neural activity.

## Introduction

The most characteristic activity of the cerebral cortex observed in human electroencephalography (EEG) at rest and during wakefulness are alpha waves [1, 2]. From the perspective of brain structure, alpha wave fluctuations, oscillations, have been thought to occur as cortical rhythms,

(Miltiadous A, Tzimourta KD, Afrantou T, Ioannidis P, Grigoriadis N, Tsalikakis DG, Angelidis P, Tsipouras MG, Glavas E, Giannakeas N, Tzallas AT. A Dataset of Scalp EEG Recordings of Alzheimer's Disease, Frontotemporal Dementia and Healthy Subjects from Routine EEG. Data, 202b;8(6):95. doi: 10.390/data8060095).

**Funding:** Minoru Hoshiyama reports financial support was provided by JSPS Grant-in-Aid for Scientific Research (C) (20K07881), and Minoru Hoshiyama and Hitoshi Hirata were financially supported by Japan Agency for Medical Research and Development (AMED) (AMED-CREST: 23gm1510005h0003). The authors declare that they have no known competing financial interests or personal relationships that could have potentially to influence the work reported in this article. There was no additional external funding received for this study. The above funders had no role in the study design, data collection and analysis, decision to publish, or preparation of the manuscript.

**Competing interests:** The authors have declared that no competing interests exist.

or as part of a neural circuit that forms between cortices, or outside the cortex, such as in the thalamus [3–9]. Apart from the mechanism of generation, alpha wave oscillations have been variously reported to be associated with cognitive mechanisms and diseases [10–12].

In addition to the alpha oscillation itself, the amplitude of the alpha wave fluctuates, described as waxing and waning [13]. Previous researches have focused on this phenomenon [6, 14–17], but little is known about the mechanism and clinical significance of the waxing and waning of the alpha rhythm, especially in the study of EEG in patients. Although the meaning of alpha oscillations and their waxing and waning has not yet been elucidated, it has been strongly suggested that they were activities related to neural information transmission [7, 8, 18].

The rhythmic function of alpha oscillations has been suggested to have inhibitory effects on the cortex [19–21]. On the other hand, the excitatory function of alpha oscillations has also been reported, suggesting that alpha oscillations modulate the attenuation and enhancement of resting-state activity, and that alpha oscillations acted as a pacemaker for the alternation of inhibition and excitation [22]. One hypothesis concerning the timing of the inhibitory and excitatory effects of alpha oscillations, including the waxing and waning phenomenon, proposed that the rhythmicity of alpha waves mediates inhibition [19–21, 23]. Another suggested that waxing and waning resulted from the summation of multiple periodic phenomena [14]. On the other hand, Lombardi et al. [22] proposed that the waxing and waning of alpha waves in the awake resting state involved the short-term attenuation and amplification in a few alpha cycles, a few hundred milliseconds, depending on the amplitude-dependent manner [24, 25], at each peak of alpha waves [22].

The aim of the present study was to show characteristics of alpha wave fluctuation at rest, i.e. 1) the ratio of increasing and decreasing forces that form the alpha oscillation and its relationship with amplitude, and 2) the difference of these values between healthy participants and patients with dementia in whom cognitive dysfunction occurred (Fig 1). These aims were based on previous work suggesting that the waxing and waning of alpha waves is regulated in an amplitude-dependent manner [24, 25] and that this functional regulation may be disrupted in patients with dementia [26–29]. The present study assessed the moment-to-moment amplitude fluctuations rather than the periodicity of the oscillation of alpha waves. In this respect, our measurements were based on the excitatory and inhibitory functions of alpha waves reported by Lombardi et al. [22].

For the mechanisms generating alpha oscillations observed in the EEG, mathematical models, neural mass models (NMMs), have been the leading elementary models to explain the dynamics of EEG signals, from spontaneous activity to the network level [30–35]. The NMMs had properties of nonlinear oscillator models and corresponded to physiological functions at the neuronal population level [30–35]. A nonlinear oscillator model involves second-order differential equations in time, and the second-order differential equations in the NMM developed by Umehara et al. [36] well incorporated the regulation of excitatory and inhibitory neural populations.

In the present analysis, we focused on the second-order derivative with respect to time, i.e. the acceleration, which formed the envelope of the clinical EEG. We obtained the sum of the positive and negative second-order derivative values and their ratios for each of the 60 seconds of EEG. We observed whether acceleration, as a variable component of the alpha wave oscillation, differed between healthy subjects and patients as a factor involved in the envelope of the alpha wave.

Although our analysis methods were applied to conventional EEG recordings, since our analysis method involved a novel idea, we report the present results using an open EEG dataset to facilitate its validation [37–39]. The dataset contained the eyes-closed resting-state EEG recordings from a total of 88 participants, including 36 participants with Alzheimer's disease

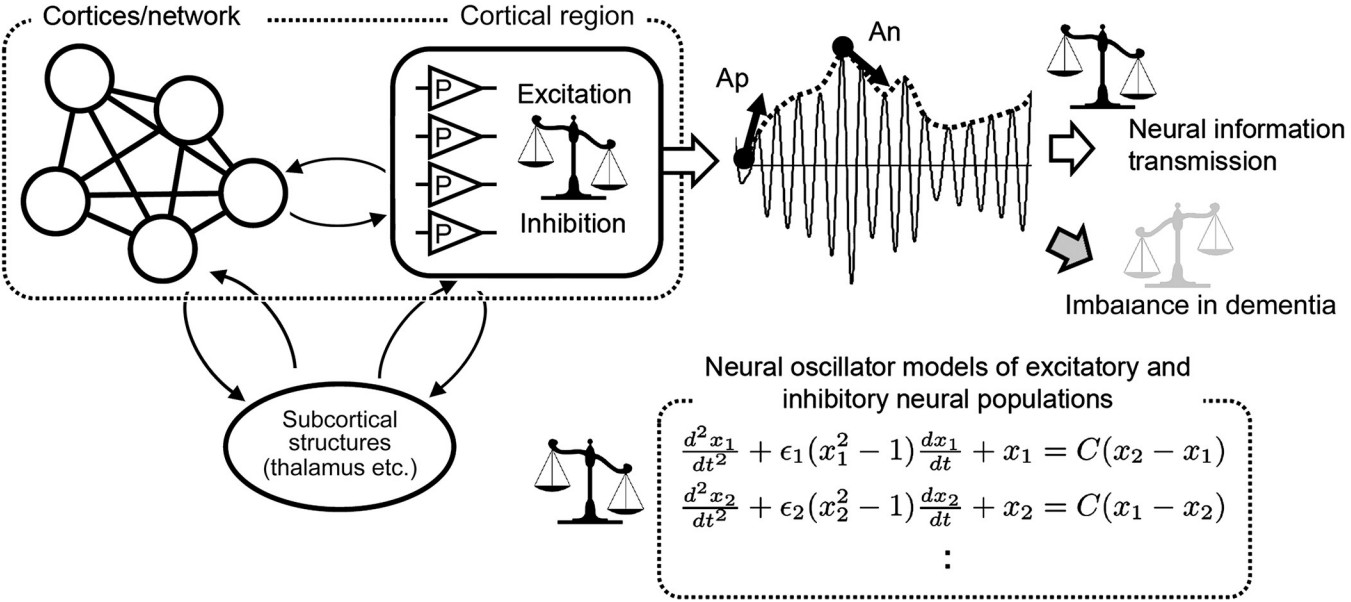

**Fig 1. Illustration of the concept of the present study.** Left: Alpha wave oscillation in a cortical area forms a balance between excitation and inhibition via neural circuits with subcortical structures such as the thalamus and other cortices and networks at rest. P: cortical pyramidal cells. Top right: The waxing and waning of alpha oscillations occurs in a balance in relation to the excitation and inhibition balance of the cortical region and is transmitted as neural information. In dementia, this balance is abnormal and functional impairment occurs. Arrows indicate positive (Ap) and negative (An) accelerations, second-order derivatives in time, of the alpha envelope. Bottom right: Mathematical implication. Neural oscillator models, such as neural mass models, contain non-linear equations of oscillation. The second-order derivatives in the equations can be inhibitory or excitatory forces, the balance of which is an important element of neural function. The small black and gray balance diagram in the figure shows a balanced and imbalanced condition, respectively. We calculated the ratio of positive and negative acceleration values, Ap-An ratio, during an analysis period in the present study. We assumed that the ratio was an indicator of the balance between excitation and inhibition in the cortical region. Note: The equations are simple oscillator model equations for illustrative purposes, not equations describing the envelope of the alpha oscillation.

(AD), 23 with frontotemporal dementia (FTD), and 29 healthy participants (normal control, NC). We compared the values obtained from our analysis and compared between the groups. In this study, the EEG of patients with dementia as well as that of normal participants was investigated. In addition to cognitive decline, EEG abnormalities have been reported in patients with dementia, such as disinhibition of alpha rhythm and cortical activity in participants with AD in previous studies [26–29]. The envelope analysis in this study was related to the parameters of excitatory and inhibitory control of brain activity, i.e., the variability characteristics of the envelope with respect to inhibitory and excitatory regulation of brain function [22, 36], and was expected to be able to detect the destruction of alpha activity in patients with dementia.

The term excitation-inhibition (E-I or E/I) balance, which has received much interest recently [40–42], has been used in a broad sense, from the molecular to the system level [43]. The present study assessed positive (acceleration positive, Ap value), negative (acceleration negative, An value) acceleration values and their ratio, Ap-An ratio, of the alpha wave envelope, and it might be difficult to directly approach their origin in neural mechanisms, as well as their relationship to the E-I balance used in other studies. However, we thought that the present study proposed a functional factor of the EEG related to the E-I regulation, since the alpha oscillations at rest mediated inhibitory and excitatory functions in the models [22, 36]. This study was believed that the present analysis would also allow us to detect the destruction of the function of alpha activity in various disorders.

## Methods

The EEG data sets used in the present study were obtained from an open dataset [37–39]. We chose to use this open dataset because it provided stable EEG recordings during resting, awake, and closed-eye conditions in the sitting position with sufficient recording time, more than 10 minutes for each participant [37–39]. Closed-eye recording was advantageous not only to avoid blink artifacts and brain activity associated with visual stimuli and eye movements, but also to maintain a constant resting state, especially in patients with dementia. The seated position was valuable in preventing participants from falling asleep.

### Participants

The dataset included resting EEG with eyes closed from 88 participants, 36 participants with AD (24 females and 12 males, mean age; 66.4 ±2.6 (SD)), 23 with FTD (9 and 14, 63.6 ±8.2), and 29 with NC (11 and 18, 67.9 ±5.4). Information of all participants was provided in S1 Table in the Supporting Information and the dataset [37–39]. A score on the Mini-Mental State Examination (MMSE) [44], a questionnaire used to assess the cognitive severity of dementia, was also provided for each participant in the data set, and the mean MMSE score was 17.8 ±4.5 (range; 9–23) and 22.2 ±2.6 (18–27) for the AD and FTD groups, respectively.

### EEG datasets

The datasets included the EEG signals obtained from 19 electrodes, Fp1, Fp2, F7, F3, Fz, F4, F8, T3, C3, Cz, C4, T4, T5, P3, Pz, P4, T6, O1, and O2, according to the international 10–20 system, with 2 reference electrodes (A1 and A2) placed on the mastoids. The inter-electrode impedance was less than 5 kΩ. Each recording was made with participants in the sitting position with eyes closed. The sampling rate was 500 Hz, and the EEG signals were collected with a time constant of 0.3 seconds and a high-cut filter at 70 Hz [37–39]. The pre-processing that had already been done on the datasets were: 1) a Butterworth bandpass filter 0. 5–45 Hz, 2) the signals were referenced to the two connected mastoid electrodes, A1-A2, 3) the Artifact Subspace Reconstruction (ASR) routine, an EEG artifact correction method included in the EEGLab/Matlab software, was applied, 4) the Independent Component Analysis (ICA) method was performed, transforming the 19 EEG signals into 19 ICA components, and ICA components categorized as eye artifacts or jaw artifacts by the EEGLAB platform were automatically excluded [37–39].

### Selection of epochs for the present analysis

The original datasets contained more than 10 minutes for each participant. We extracted 60 seconds of EEG signals for the present study. Although preprocessing and artifact rejection algorithms for electrocardiogram and eye blink artifacts were applied to the datasets, some electrical artifacts were present, as mentioned by Miltiadous et al. [37–39], possibly due to extra-brain electrical noise or some motion artifact. Therefore, we manually selected a 60-second window without significant noise for each participant. During the manual selection of the 60-second epoch, we selected the 60-second epochs based on the following criteria: 1) electromyography (EMG) burst, 2) square-wave noise, 3) abrupt baseline shift, and 4) noise greater than 300 μV. Even if the amplitude was less than 300 μV, the waves that were considered electrical artifacts because of their shape were omitted from the epochs. The epochs selected for each participant were listed in S1 Table in the Supporting Information.

It has been reported that the envelope of alpha waves contained fluctuations with periods ranging from a few seconds to tens of seconds [14]. This study was not a frequency analysis of

the envelope, but an analysis of the increase or decrease in amplitude produced by the peaks of alpha waves. As shown in Fig 2, a few peaks of the alpha envelope were obtained in 1 second, and more than several dozen peaks of the envelope were included in a 60-second period, which was considered sufficient for the present analysis. A practical reason was that even in the 10-min resting EEG recordings of each participant, there were only one or two 60-second epochs with low artifacts such as motion. Not all participants had multiple suitable continuous 60-second recordings for analysis, and we had to select a 60-second epoch from the 10-min recording, particularly for participants with dementia (S1 Table, Supporting Information).

## Data analysis (Figs 2 and 3)

We used MATLAB® (MathWorks®, US) and the MATLAB®-based open-source software Brainstorm for the following analysis [45]. First, bandpass filtering between 8 and 12 Hz was applied to obtain alpha band EEG activity for 60 seconds. The peak frequency in the alpha frequency band in the power spectrum density was calculated at each electrode in each participant using Welch's method, which divided the signals into 50% overlapping windows for each second.

The baseline was offset by the signal level averaged over 60 seconds, and the signal was rectified by taking the absolute value of the sampling points. In the present study, we analyzed the amplitude of the envelope formed by the alpha wave peaks. The envelope could be plotted for positive and negative alpha wave peaks relative to the baseline. Since alpha waves have been observed as oscillations on scalp EEG [46], the positive and negative deviations of alpha waves were treated as caused by similar temporal sequences of oscillations in the present study. To obtain more temporal information about the number of peaks per unit time, the waveforms were rectified to create the envelopes.

An envelope of the time course of the alpha activity at each electrode was calculated using the MATLAB command. For the rectified alpha activity, 60 seconds was used as the magnitude of the signal to create the upper envelope. The envelope was calculated using the discrete Fourier transform as implemented in the Hilbert transform. The function first removed the mean of the amplitude of the rectified alpha activity and added it back after calculating the envelope. Each point of the envelope was determined by spline interpolation over a value at each sample point. The calculation formula for generating the envelope (g(t)) from the alpha activity (f(t)) was as follows. The MATLAB "envelope ('peak')" command to create an envelope included the Hilbert transform processes.

$$h(t) = \text{sqrt} \{(f(t))\hat{2} + (H(f(t))\hat{2}\} \tag{1}$$

$$g(t) = |h(t)| \tag{2}$$

H; Hilbert transform, sqrt; square root, h(t); intermediate formula

The envelope of the alpha wave time course was second-order differentiated at each sampling point. We defined an absolute value of a positive and a negative value of the second-order differentiation at each sampling point as acceleration positive (Ap) and negative (An) values, respectively, and the Ap-An ratio was a ratio of Ap to An values at each sampling point.

$$\text{Ap value} = |\text{positive second−order differentiated value}| \tag{3}$$

$$\text{An value} = |\text{negative second−order differentiated value}| \tag{4}$$

$$\text{Ap−An ratio} = [\text{Ap value}]/[\text{An value}] \tag{5}$$

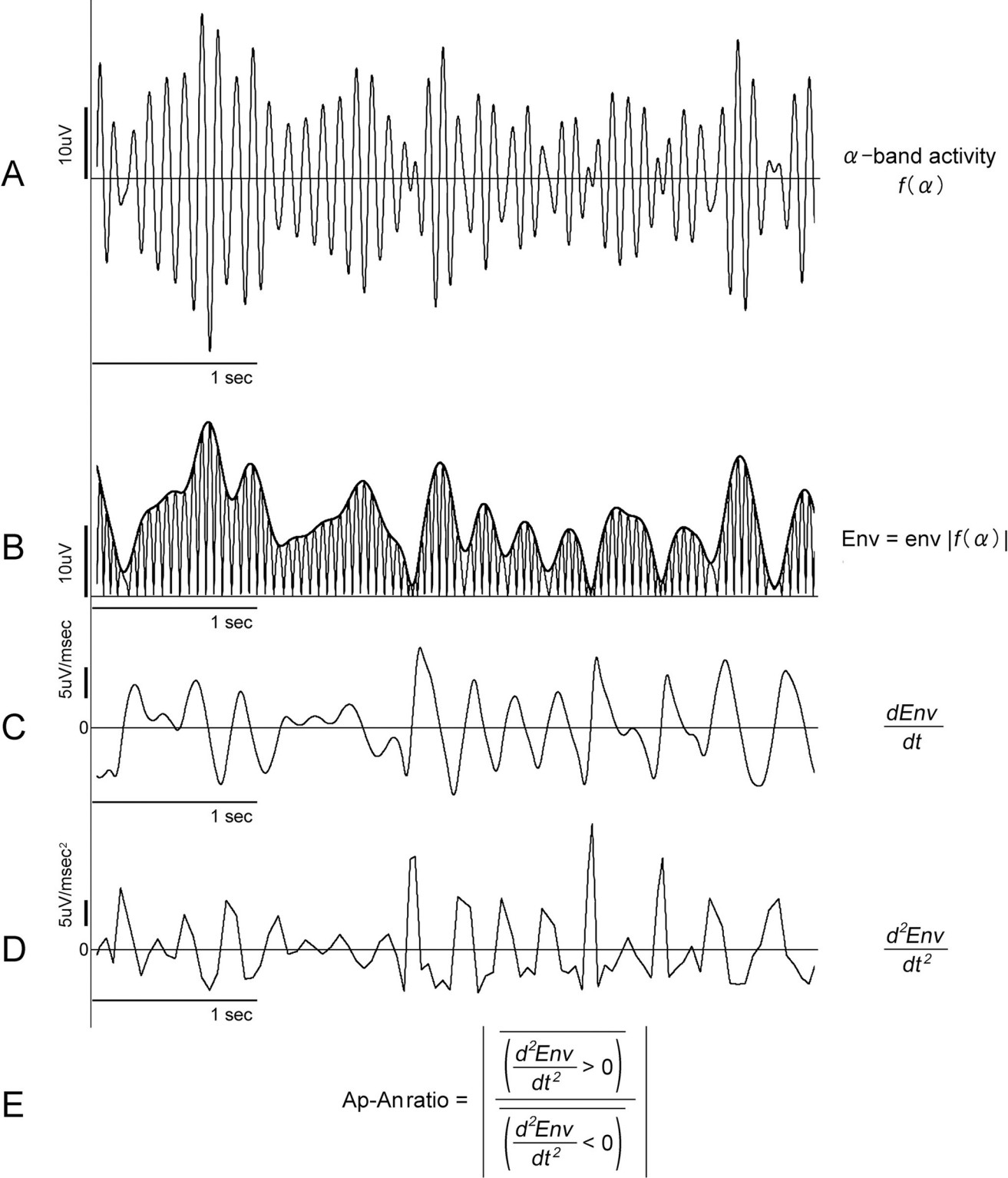

**Fig 2. Representation of the electroencephalography (EEG) signal analysis method.** The figures were created using 5-second EEG signals obtained at the F4 electrode in sub-037 (221–226 sec). A: EEG signals filtered by alpha band frequency (8–12 c/s). B: EEG signals rectified (thin line) and peak envelope (solid line). C: First derivative of the envelope. D: Second-order derivative of the envelope. E: An Ap-An ratio was an absolute value of the ratio of mean positive derivatives to mean negative derivatives for 60 seconds.

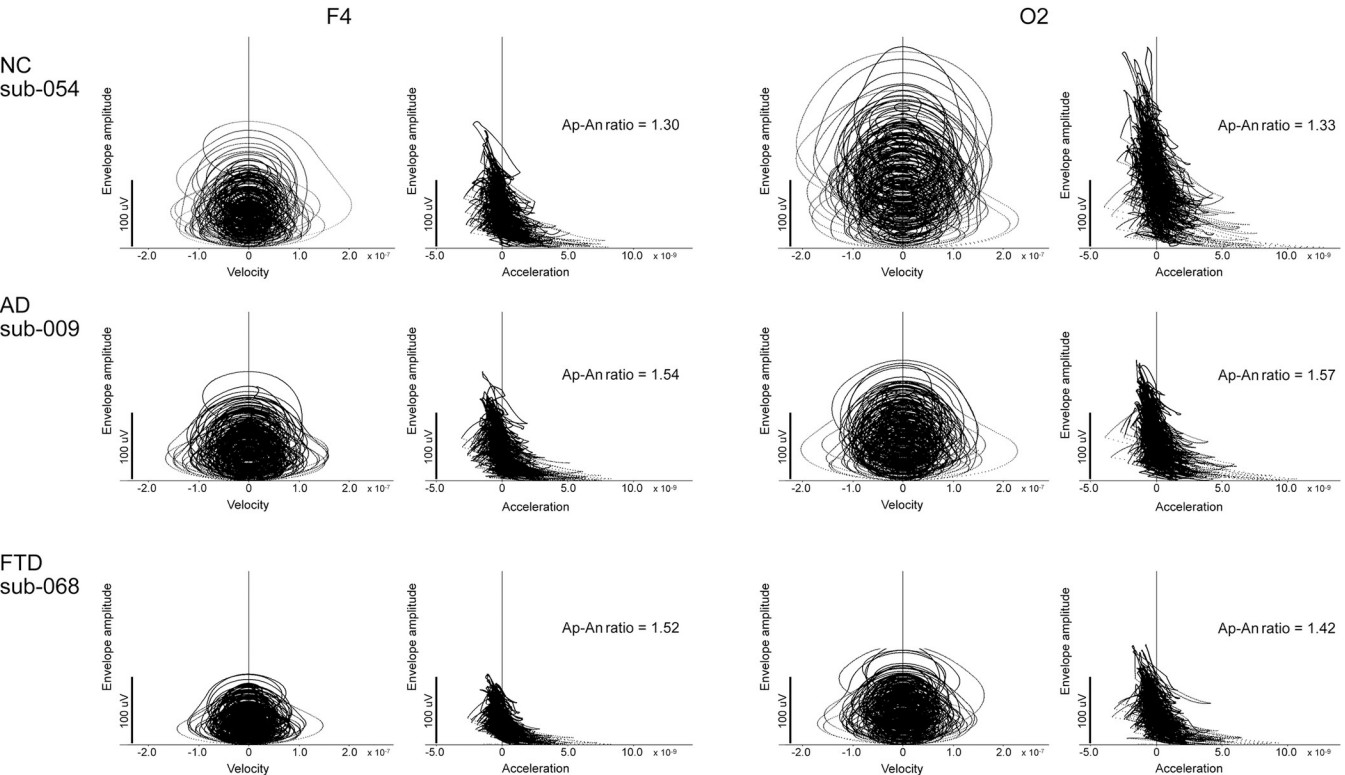

**Fig 3. Relationship between envelope amplitude and first and second-order derivatives at electrodes F4 and O2 at each sampling point in 60 seconds.**
One participant was presented in each group. Each vertical axis represents the envelope amplitude and the horizontal axis represents the first (Velocity) and the second-order (Acceleration) derivatives in time of the envelope (see Fig 2D) at each sampling point. While the first derivative values were symmetrically distributed with positive and negative values for each envelope amplitude, the second-order derivative values showed an asymmetrical distribution depending on the value of the envelope amplitude. In other words, the positive second-order derivative values were larger for smaller envelope amplitudes, while the negative second-order derivative values had a similar distribution for all amplitudes. The Ap-An ratios of the NC participant were lower than those of the AD and FTD participants, regardless of the envelope amplitude. NC: normal control; AD: participants with Alzheimer's disease; FTD: participants with frontotemporal dementia.

The average of the Ap and An values and the Ap-An ratio for 60 seconds were obtained at each electrode. The root mean square (RMS) of the alpha activity for 60 seconds at each electrode was calculated as the envelope amplitude.

## Statistical analysis

The mean envelope amplitude, peak frequency of the alpha frequency band, Ap and An values, and Ap-An ratio were calculated for all electrodes. The envelope amplitude, the Ap-An ratio, and Ap and An values were compared between groups, AD and NC, and FTD and NC groups at each electrode site.

Pearson's correlation coefficient between envelope amplitude and Ap and An values and Ap-An ratio was compared between groups at each electrode site. The correlation between envelope amplitude and Ap and An values and Ap-An ratio between electrodes was also calculated. A p-value was calculated from each correlation coefficient. Repeated measures (for 19 electrodes) analysis of variance (ANOVA) was used to compare grand means of 19 electrodes between groups.

A p-value less than 0.05 was considered significant, and the false discovery rate (FDR) or Bonferroni-Dun test was used for multiple comparisons.

## Results

Although there was no significant difference in age between the three groups, the MMSE score was lower in the AD group than in the FTD group (t-test, $p < 0.0001$). Eighty-eight sets of EEG recordings were successfully obtained from the open source. The rectified EEG signals, the upper envelope, and the values of the first and second-order derivatives of the envelope at each sampling point were obtained (Fig 2). The relationship between the value of the first and second-order derivatives and the amplitude of the envelope at each sampling point at F4 and O2 in one participant from each group was expressed in Fig 3. Asymmetry in positive and negative values was observed in the second-order derivatives. While the first derivative values were symmetrically distributed with positive and negative values for each envelope amplitude, the second-order derivative values showed an asymmetrical distribution depending on the value of the envelope amplitude. In other words, the positive second-order derivative values were larger for smaller envelope amplitudes, while the negative second-order derivative values had a relatively similar distribution to the positive second-order derivative for all amplitudes (Fig 3). The mean amplitude expressed as root mean square (RMS) of alpha activity, the peak frequency of alpha activity for 60 seconds analyzed are shown in Table 1. The peak alpha frequency in the occipital areas, at each electrode O1 and O2 where the alpha waves were typically

**Table 1. Amplitude of the envelope and peak frequency in the alpha frequency band at each electrode.**

| Electrode | Amplitude (x10 uV) mean ±SD | | | Peak frequency of alpha band mean ±SD | | |
|---|---|---|---|---|---|---|
| | NC | AD | FTD | NC | AD | FTD |
| Fp1 | 6.6 ±2.0 | 5.7 ±1.1* | 5.7 ±1.1 | 9.4 ±0.8 | 9.1 ±1.0 | 9.2 ±1.0 |
| Fp2 | 6.6 ±2.0 | 5.5 ±1.1* | 5.5 ±1.1* | 9.4 ±0.8 | 9.0 ±1.0* | 9.1 ±1.1 |
| F3 | 5.8 ±1.6 | 5.1 ±1.0* | 5.1 ±0.9 | 9.3 ±0.7 | 9.0 ±0.9 | 9.3 ±1.0 |
| F4 | 5.9 ±1.7 | 4.9 ±0.7* | 4.9 ±0.7* | 9.4 ±0.8 | 8.8 ±0.7* | 9.2 ±1.0 |
| C3 | 4.9 ±1.0 | 4.6 ±0.5 | 4.6 ±0.5 | 9.1 ±0.6 | 8.8 ±0.8 | 9.2 ±0.9 |
| C4 | 4.9 ±1.0 | 4.6 ±0.5 | 4.6 ±0.5 | 9.3 ±0.7 | 8.8 ±0.8* | 9.1 ±0.9 |
| P3 | 7.5 ±2.8 | 5.7 ±1.7* | 5.7 ±1.7* | 9.7 ±0.8 | 9.1 ±8.7* | 9.2 ±1.0 |
| P4 | 7.0 ±2.3 | 6.0 ±2.1* | 6.0 ±2.1 | 9.6 ±8.7 | 9.1 ±8.7* | 9.3 ±1.0 |
| O1 | 10.5 ±4.3 | 7.5 ±3.8* | 7.5 ±3.8* | 9.6 ±0.9 | 9.4 ±0.9 | 9.3 ±1.0 |
| O2 | 11.2 ±4.1 | 7.9 ±4.0* | 7.9 ±4.0* | 9.6 ±0.9 | 9.2 ±0.8* | 9.5 ±1.1 |
| F7 | 6.3 ±1.9 | 5.7 ±1.4 | 5.7 ±1.4 | 9.2 ±6.4 | 8.8 ±0.8* | 9.2 ±0.9 |
| F8 | 6.4 ±1.9 | 5.6 ±1.2* | 5.6 ±1.2 | 9.4 ±0.8 | 8.8 ±0.8* | 9.3 ±0.9 |
| T3 | 6.4 ±1.8 | 5.8 ±1.6* | 5.8 ±1.6 | 9.3 ±0.7 | 9.0 ±0.8 | 9.2 ±0.9 |
| T4 | 6.1 ±1.5 | 5.8 ±1.4 | 5.8 ±1.4 | 9.3 ±0.7 | 8.8 ±0.8* | 9.1 ±1.0 |
| T5 | 10.1 ±4.2 | 7.1 ±2.7* | 7.0 ±2.7* | 9.6 ±0.9 | 9.2 ±8.4* | 9.2 ±1.0 |
| T6 | 9.3 ±3.2 | 7.7 ±3.2* | 7.7 ±3.2 | 9.4 ±0.9 | 9.1 ±0.7 | 9.4 ±1.1 |
| Fz | 5.9 ±1.8 | 5.0 ±0.9* | 5.0 ±9.0* | 9.3 ±0.8 | 8.9 ±0.8* | 9.3 ±1.0 |
| Cz | 4.8 ±8.4 | 4.6 ±0.8 | 4.6 ±8.1 | 9.1 ±0.6 | 8.7 ±0.6* | 9.1 ±0.9 |
| Pz | 7.1 ±2.6 | 5.7 ±2.0* | 5.7 ±2.0* | 9.5 ±0.8 | 9.0 ±0.9* | 9.4 ±0.9 |
| All electrodes | 7.0 ±2.0 | 5.2 ±6.4** | 5.8 ±1.0** | 9.4 ±0.2 | 9.0 ±0.2** | 9.2 ±0.2** |

Amplitude: Envelope amplitude was calculated as the root mean square.

NC: normal control; AD: participants with Alzheimer's disease; FTD: participants with frontotemporal dementia.

Gray columns indicate significant difference between NC ((p < 0.05, t-test, unconditioned

*p < 0.05, conditioned by FDR

**p < 0.05, Bonferroni-Dun test).

**Table 2. The Ap and An values of the alpha envelope and the Ap-An ratio at each electrode.**

| Electrode | Ap value (x e-10) mean ±SD | | | An value (x e-10) mean ±SD | | | Ap-An ratio mean ±SD | | |
|---|---|---|---|---|---|---|---|---|---|
| | NC | AD | FTD | NC | AD | FTD | NC | AD | FTD |
| Fp1 | 7.1 ±1.1 | 7.2 ±1.2 | 7.3 ±1.7 | 5.1 ±1.3 | 5.2 ±0.8 | 5.1 ±1.1 | 1.38 ±0.09 | 1.39 ±0.05 | 1.41 ±0.06 |
| Fp2 | 7.0 ±1.1 | 7.2 ±1.2 | 7.0 ±1.4 | 5.1 ±1.4 | 5.1 ±0.8 | 5.1 ±0.9 | 1.40 ±0.07 | 1.39 ±0.05 | 1.39 ±0.06 |
| F3 | 6.5 ±1.0 | 6.7 ±0.9 | 6.5 ±1.2 | 4.6 ±1.3 | 4.7 ±0.6 | 4.6 ±0.7 | 1.40 ±0.06 | 1.42 ±0.05 | 1.40 ±0.06 |
| F4 | 6.5 ±0.9 | 6.6 ±1.0 | 6.5 ±1.0 | 4.6 ±1.2 | 4.7 ±0.6 | 4.6 ±0.7 | 1.40 ±0.06 | 1.41 ±0.05 | 1.42 ±0.05 |
| C3 | 6.0 ±0.7 | 6.2 ±0.8 | 6.3 ±0.6 | 4.3 ±1.2 | 4.3 ±0.5 | 4.2 ±0.4 | 1.40 ±0.05 | 1.42 ±0.04 | 1.42 ±0.06 |
| C4 | 6.1 ±0.9 | 6.1 ±0.8 | 6.3 ±0.8 | 4.3 ±1.2 | 4.4 ±0.5 | 4.4 ±0.6 | 1.41 ±0.05 | 1.41 ±0.04 | 1.43 ±0.05 |
| P3 | 6.9 ±1.4 | 6.7 ±1.1 | 6.6 ±1.1 | 5.0 ±1.4 | 4.8 ±0.7 | 4.7 ±0.8 | 1.37 ±0.08 | 1.40 ±0.06 | 1.41 ±0.09 |
| P4 | 6.7 ±1.3 | 6.8 ±1.2 | 6.8 ±1.3 | 4.9 ±1.6 | 4.8 ±0.8 | 4.8 ±0.9 | 1.38 ±0.07 | 1.42 ±0.07 | 1.41 ±0.09 |
| O1 | 7.8 ±2.0 | 7.5 ±1.5 | 7.3 ±1.7 | 5.7 ±1.8 | 5.3 ±1.0 | 5.3 ±1.2 | 1.35 ±0.08 | 1.41 ±0.08* | 1.38 ±0.10 |
| O2 | 8.1 ±2.1 | 7.7 ±1.7 | 7.5 ±1.8 | 6.0 ±1.8 | 5.4 ±1.1 | 5.4 ±1.2 | 1.35 ±0.09 | 1.41 ±0.07* | 1.39 ±0.10 |
| F7 | 6.9 ±1.2 | 7.4 ±1.3 | 7.2 ±1.6 | 5.0 ±1.4 | 5.2 ±0.9 | 5.1 ±1.2 | 1.39 ±0.06 | 1.41 ±0.05 | 1.40 ±0.04 |
| F8 | 7.0 ±1.2 | 7.3 ±1.2 | 7.1 ±1.5 | 5.0 ±1.4 | 5.2 ±0.8 | 5.1 ±1.1 | 1.39 ±0.05 | 1.41 ±0.05 | 1.39 ±0.06 |
| T3 | 7.1 ±1.1 | 7.2 ±1.2 | 7.2 ±1.4 | 5.1 ±1.4 | 5.0 ±0.8 | 5.1 ±1.1 | 1.38 ±0.04 | 1.42 ±0.05* | 1.41 ±0.05 |
| T4 | 6.9 ±1.2 | 7.1 ±1.5 | 7.1 ±1.4 | 5.0 ±1.3 | 5.0 ±0.9 | 5.1 ±1.0 | 1.39 ±0.05 | 1.41 ±0.05 | 1.40 ±0.07 |
| T5 | 7.6 ±2.0 | 7.3 ±1.4 | 7.2 ±1.4 | 5.7 ±1.9 | 5.3 ±1.0 | 5.2 ±1.1 | 1.33 ±0.09 | 1.38 ±0.05* | 1.38 ±0.07 |
| T6 | 7.5 ±1.6 | 7.4 ±1.5 | 7.6 ±2.0 | 5.5 ±1.5 | 5.3 ±1.1 | 5.5 ±1.4 | 1.35 ±0.07 | 1.39 ±0.06* | 1.39 ±0.10 |
| Fz | 6.3 ±0.9 | 6.5 ±1.1 | 6.4 ±1.1 | 4.6 ±1.2 | 4.6 ±0.7 | 4.5 ±0.7 | 1.38 ±0.06 | 1.40 ±0.05 | 1.40 ±0.06 |
| Cz | 5.9 ±0.8 | 6.0 ±0.8 | 6.2 ±0.8 | 4.2 ±1.1 | 4.3 ±0.5 | 4.6 ±0.6 | 1.40 ±0.04 | 1.40 ±0.05 | 1.42 ±0.06 |
| Pz | 6.6 ±1.3 | 6.5 ±1.2 | 6.3 ±1.0 | 4.8 ±1.3 | 4.6 ±0.7 | 4.6 ±0.7 | 1.38 ±0.07 | 1.41 ±0.04 | 1.39 ±0.09 |
| All electrodes | 6.8 ±6.0 | 6.9 ±4.9** | 6.8 ±4.9** | 5.0 ±5.0 | 4.9 ±3.6** | 4.8 ±3.8** | 1.38 ±0.02 | 1.41 ±0.01** | 1.40 ±0.02** |

NC: normal control; AD: participants with Alzheimer's disease; FTD: participants with frontotemporal dementia. All electrodes: grand mean of values for 19 electrodes. Gray columns indicate significant difference between NC and AD/FTD groups (p < 0.05, t-test, unconditioned, *p < 0.05, conditioned by FDR, **p < 0.05, Bonferroni-Dun test)

found [47], for each participant was shown in S1 Table in the Supporting Information. The Ap and An values and the Ap-An ratio for 60 seconds in each group are shown in Table 2.

## Signal analysis in normal participants

Twenty-nine normal participants were tested first. For 19 electrodes, the Ap-An ratio (r = -0.921, p < 0.000001) was strongly negatively correlated with the mean envelope amplitude (Fig 4), and the Ap (0.920, p < 0.000001) and An (0.945, p < 0.000001) values were also strongly positively correlated with the mean envelope amplitude. For each electrode, the correlations between Ap-An ratio and mean envelope amplitude are shown in Table 3. There were significant correlations between Ap-An ratio and mean envelope amplitude at some electrodes from frontal to occipital areas in the NC group (Table 3, Fig 5). There was an asymmetry of the correlation in the frontal, central and parietal areas (Table 3, Fig 5).

## Signal analysis in participants with dementia

Thirty-six participants with AD and 23 participants with FTD were successfully analyzed. Mean envelope amplitude, peak frequency in the alpha frequency band (Table 1), Ap-An ratio, Ap and An values (Table 2) at each electrode were presented. In the AD group, among 19 electrodes, Ap-An ratio (r = -0.306) was not correlated with envelope amplitude (Table 3), but Ap (0.946, p < 0.000001) and N (0.948, p < 0.000001) values were strongly correlated with

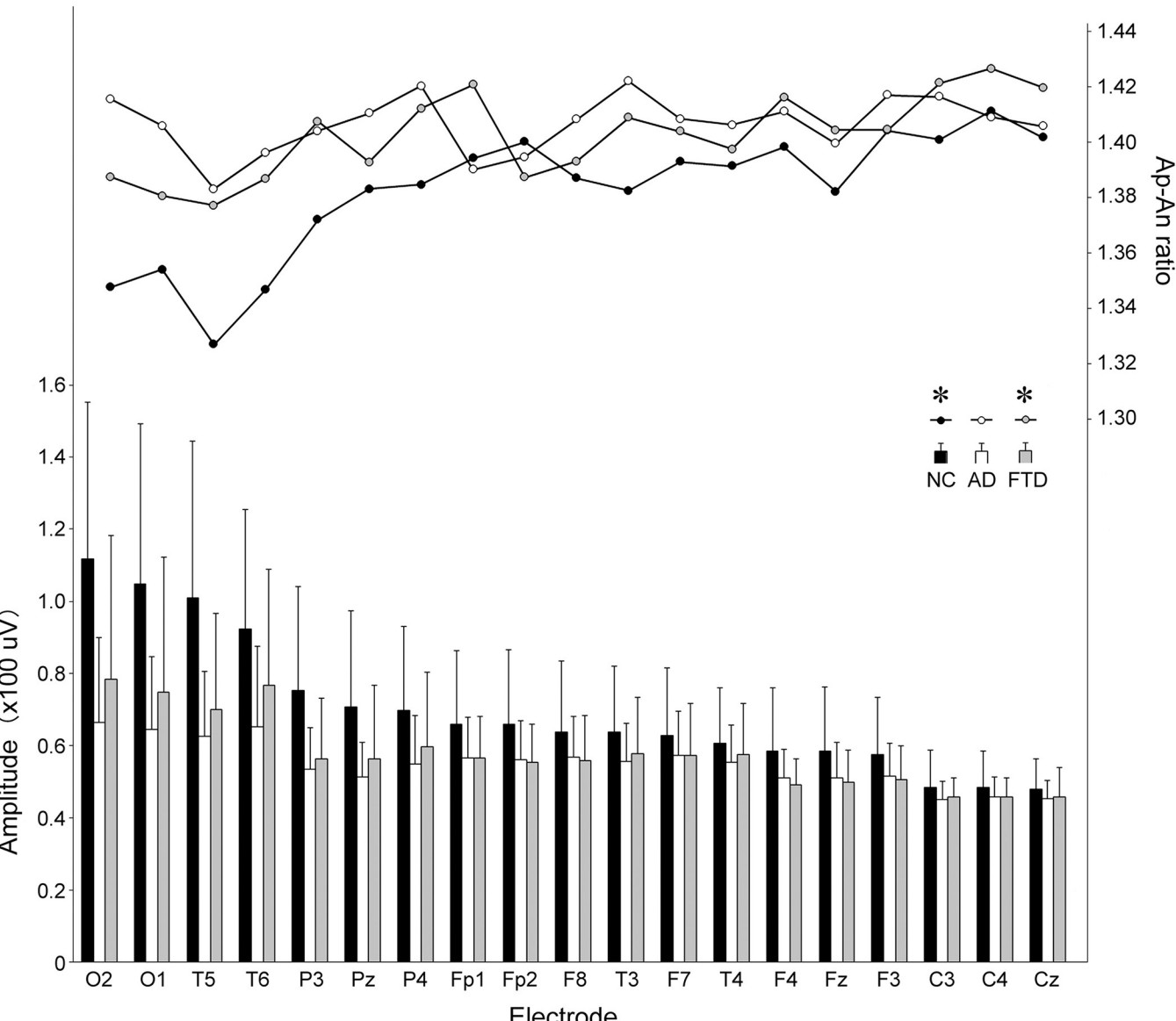

**Fig 4. The mean envelope amplitude and mean Ap-An ratio at each electrode in the groups.** The graphs are arranged from left to right, starting with the electrode with the highest value in the NC group. The Ap-An ratio was correlated with the envelope amplitude as shown in Table 3, but the order of the Ap-An ratio did not necessarily coincide with the order of the envelope amplitude. Black columns/circles (NC): normal control; white columns/circles (AD): participants with Alzheimer's disease; and light gray columns/circles (FTD): participants with frontotemporal dementia. A vertical bar indicates a standard deviation for each envelope amplitude. The Ap-An ratio was strongly negatively correlated with envelope amplitude across 19 electrodes in the NC (r = -0.921, *p < 0.000001) and FTD (-0.771, *0.00011) groups, but not in the AD group (r = -0.306).

envelope amplitude. In the FTD group, Ap-An (-0.771, 0.00011) (Table 3), E (0.734, $p < 0.00001$), and I (0.869, $p < 0.00001$) values were strongly correlated with envelope amplitude among 19 electrodes.

The mean Ap-An ratio for 19 electrodes in the AD (1.41 ±0.01 (SD)) and FTD (1.40 ±0.02) groups showed a greater Ap-An ratio than in the NC group (1.38 ±0.02, p<0.05). When comparing the mean for each electrode with a repeated measures ANOVA with Bonferroni-Dun test, the Ap-An ratio in the AD and FTD groups was greater than that in the NC participants with significance ($p < 0.0001$) (Table 2). Both the grand mean Ap and An values were greater

**Table 3. Correlation between alpha envelope amplitude and the Ap-An ratio at each electrode.**

| Electrode | Correlation coefficient Between Amplitude and Ap-An ratio | | | p value | | |
|---|---|---|---|---|---|---|
| | NC | AD | FTD | NC | AD | FTD |
| Fp1 | -0.532 | -0.249 | -0.078 | 0.001* | 0.142 | 0.724 |
| Fp2 | -0.550 | 0.027 | -0.181 | 0.001* | 0.877 | 0.409 |
| F3 | -0.413 | -0.080 | -0.324 | 0.014* | 0.644 | 0.132 |
| F4 | -0.568 | 0.088 | -0.247 | 0.001* | 0.611 | 0.256 |
| C3 | -0.513 | 0.073 | 0.092 | 0.002* | 0.673 | 0.675 |
| C4 | 0.018 | 0.162 | -0.242 | 0.918 | 0.346 | 0.266 |
| P3 | -0.604 | -0.232 | -0.596 | 0.0002* | 0.173 | 0.003* |
| P4 | -0.464 | -0.040 | -0.647 | 0.005* | 0.819 | 0.001* |
| O1 | -0.473 | -0.498 | -0.623 | 0.005* | 0.002* | 0.001* |
| O2 | -0.338 | -0.350 | -0.617 | 0.047 | 0.036 | 0.002* |
| F7 | -0.442 | -0.027 | -0.138 | 0.008* | 0.877 | 0.529 |
| F8 | -0.434 | -0.143 | -0.274 | 0.010* | 0.407 | 0.206 |
| T3 | -0.467 | 0.049 | -0.519 | 0.005* | 0.778 | 0.011* |
| T4 | -0.416 | 0.129 | -0.428 | 0.014* | 0.452 | 0.042 |
| T5 | -0.455 | -0.346 | -0.676 | 0.006* | 0.039 | 0.001* |
| T6 | -0.427 | -0.540 | -0.394 | 0.011* | 0.001* | 0.063 |
| Fz | -0.396 | -0.026 | -0.422 | 0.019* | 0.878 | 0.045 |
| Cz | -0.104 | 0.139 | -0.189 | 0.549 | 0.419 | 0.387 |
| Pz | -0.369 | 0.056 | -0.718 | 0.029* | 0.745 | 0.001* |
| All electrodes | -0.921 | -0.306** | -0.771 | 0.0001** | 0.202 | 0.0001** |

NC: normal control, AD: participants with Alzheimer's disease, FTD: participants with frontotemporal dementia.

Pearson's correlation coefficient: Gray columns indicate significant difference between the NC and AD/FTD groups (($p < 0.05$, t-test, unconditioned

**$p < 0.05$, Bonferroni-Dun test). All electrodes: grand mean of values for 19 electrodes. Ap value: Gray columns indicate significant correlation between alpha envelope amplitude and Ap-An ratio ($p < 0.05$, t-test, unconditioned

*$p < 0.05$, conditioned by FDR, **$p < 0.05$, Bonferroni-Dun test).

in the AD and FTD groups than in the NC group ($p < 0.0001$, repeated measures ANOVA with Bonferroni-Dun test) (Table 2).

Correlation curves between envelope amplitude and Ap-An ratio were significantly different between NC and AD groups at FP2, F4 and C3, and between NC and FTD groups at Pz (Table 3, Fig 5). The distribution of correlation curves for 19 electrodes in each group was shown in Fig 6.

## Discussion

The present study proposed a functional characteristic of the oscillation of the amplitude of alpha waves, the envelope of alpha activity. Our simple but novel analytical method found functional features of oscillations in alpha activity.

Our findings in the present study were summarized as follows: 1) the envelope of alpha activity was formed by a consistent balance between positive and negative accelerations, the Ap-An ratio, although the ratio might differ between brain regions, 2) the Ap-An ratio was negatively correlated with envelope amplitude, which might be a regulatory function of alpha activity, 3) participants with AD and FTD types of dementia showed a larger Ap-An ratio, 4) the distribution of electrodes, at which a correlation between the envelope amplitude of alpha activity and the Ap-An ratio was observed in healthy controls, was not the same in participants

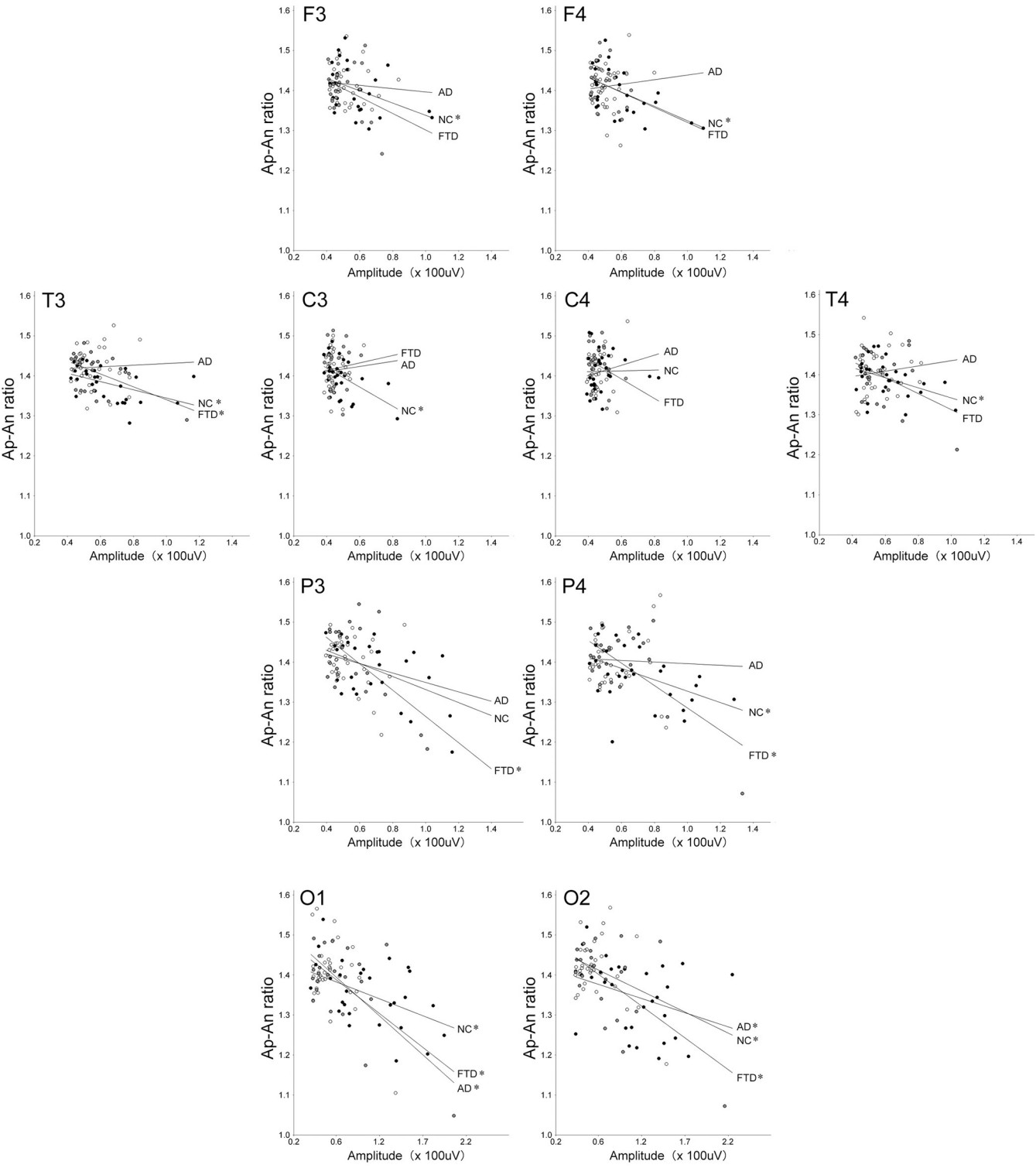

**Fig 5. Correlation between alpha envelope amplitude and Ap-An ratio at electrode (10 electrodes were shown).** NC (black circles): normal control; AD (white circles): participants with Alzheimer's disease; FTD (gray circles): participants with frontotemporal dementia. Correlation curves were shown for each electrode. Significant correlation coefficient: *p < 0.05, corrected by FDR.

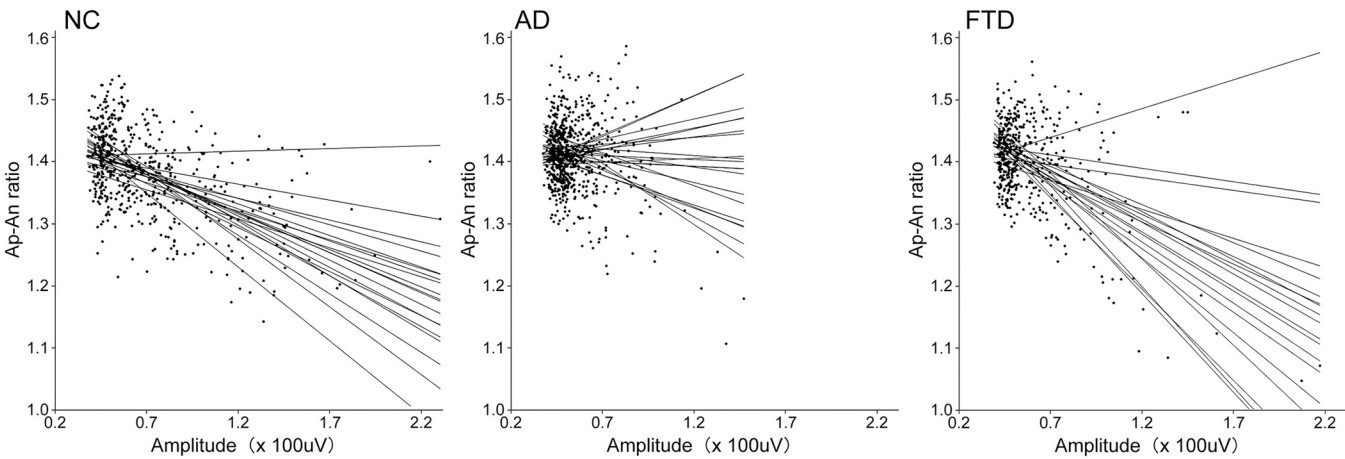

**Fig 6. Correlation between envelope amplitude (Amplitude) and the Ap-An ratio.** All values in each group were plotted as black dots, and correlation curves for 19 electrodes were shown. NC: normal control; AD: participants with Alzheimer's disease; FTD: participants with frontotemporal dementia.

with dementia. These findings suggested that alpha-envelope analysis might be a potential bio-marker to assess normal brain function and the underlying pathophysiology in various neuro-logical and psychiatric disorders involving brain dysfunction.

## Envelope of alpha activity and the Ap-An ratio in healthy participants

Since the participants in the present study were age-matched to the participants with demen-tia, we could only discuss the middle and old age of healthy participants. One of the notable findings for the envelope characteristics was that the envelope of alpha activity showed a con-sistent Ap-An retio. The Ap-An ratio was greater than 1, and the mean Ap-An ratio was approximately 1.38 for normal participants using the present analysis (Table 2), although the value should differ depending on the method of drawing the envelope. For a given envelope amplitude, the larger positive force was biased toward smaller envelope amplitudes, whereas the negative force appeared to be almost unbiased with respect to envelope amplitude, as shown in Fig 3. However, it remained unclear from the results of the present study whether the envelope amplitude was controlled by the Ap-An ratio, but it was suggested that the Ap-An ratio could be one of the markers characterizing the envelope amplitude.

Another finding was that the Ap-An ratio was negatively correlated with envelope ampli-tude in healthy participants. At a sampling point when the envelope amplitude was large, the relative negative force was greater. The amplitude of alpha waves recorded in the EEG has been considered to be the sum of multiple excitatory and inhibitory neural mechanisms, including multiple feedback processes regulated at the cortical and subcortical levels [5, 6]. Furthermore, previous studies have reported that gamma activity is associated with the inhibi-tory function of brain activity [48, 49]. Modulation of the envelope amplitude of alpha activity might lead to regulation via inhibitory neurons that generate gamma activity in subcortical and cortical structures and vice versa [7, 50]. In the present study, alpha wave fluctuations were analyzed, so the relationship with other frequencies was not considered. It should be noted that the alpha wave fluctuations were not only determined by the alpha waves them-selves, but also by functions of other EEG frequencies [7, 50].

Because of the multiple factors involved in the excitation and suppression of the alpha oscil-lation, it was not possible to distinguish from the results of the present study whether the Ap-An ratio changed secondarily with the envelope amplitude or whether it was the cause of the

envelope amplitude generation. However, based on the model proposed by Lombardi et al. (2023) [22], an excitatory or inhibitory function of alpha activity occurred with each peak of alpha activity, depending on the amplitude, e.g., high-amplitude alpha bursts acted as inhibitory effects. Although the present Ap-An ratio might be an indirect and macroscopic phenomenon, it could be a parameter that lead to the excitatory and inhibitory function of alpha activity. Since the alpha wave activity obtained at the scalp has been considered as the sum of modifications by different neuronal activities at the subcortical and cortical levels [4–9, 30], the correspondence with the neuronal and synaptic levels could not be discussed from the present results. However, the difference in the Ap-An ratio between the electrodes, as shown in Fig 4 suggested that there might be regional characteristics of the Ap-An ratio in relation to the envelope amplitude and the function of alpha waves.

## Envelope of alpha activity and the Ap-An ratio in participants with dementia

We chose the EEG datasets provided by Miltiadous et al. [37–39] and compared the Ap-An ratio between healthy participants and participants with dementia, in which the alpha oscillations associated with the excitatory and inhibitory functions might be destructed [26–29, 51]. The present results showed a larger Ap-An ratio in the participants with dementia than in the healthy participants, suggesting that there were brain regions where the control of alpha wave amplitude differed between normal participants and patients with dementia. In the previous studies, using transcranial magnetic stimulation [26], an invasive method [29] and network analysis [27, 28, 48], showed disinhibition of cortical activity in the patients with dementia, and Martínez-Cañada et al. [52] reported the hyperexcitability in patients with AD and mild cognitive impairment (MCI) by analyzing their excitation/inhibition ratio from the power spectrum and 1/f slopes of resting state EEG and magnetoencephalography (MEG). Given that amplification of alpha waves caused an inhibitory function [19–21], the greater Ap-An ratio in dementia patients than in normal participants, which induced an increase in alpha waves, could result in a compensatory phenomenon that attempted to increase the alpha waves, although it might be unable to increase alpha wave activity in the degenerated brain. The present method of calculating the Ap-An ratio made it possible to assess the functional state of the alpha waves at each electrode or in each brain region in order to compare the regional characteristics or pathology in different diseases, as shown below in the cases of FTD.

In the FTD group, the destruction of the Ap-An ratio was limited to the electrodes in the temporal region, whereas in the AD group, the Ap-An ratio was greater than that of the NC group from the parietal to the occipital region in addition to the temporal region (Table 2). When the correlation between envelope amplitude and Ap-An ratio was compared between the NC and FTD groups, it was lost in frontal and the temporal (T6) electrode areas in the FTD group, and correlated only in occipital and temporal (T5 and T6) areas in AD (Table 3). We assumed that the difference in electrode areas showing destruction of the Ap-An ratio would depend on the type and progression of the disease. Since the MMSE score was higher in the FTD group than in the AD group, the difference in the distribution of the Ap-An ratio could simply be related to the severity of the dementia, as shown between the MCI and AD groups in the previous study [52]. At the same time, however, we thought that the disruption of the Ap-An ratio might also characterize the type of dementia. Occipital areas were relatively preserved in morphology in AD [53], and frontal atrophy was one of the features in FTD [54, 55]. Although 19-channel EEG recording could not determine a specific relationship between an electrode and a brain region, electrodes with loss of significant correlation between envelope amplitude and the Ap-An ratio were not inconsistent with the pathologies in AD and

FTD. We had to mention that the MMSE scores were different, higher in the FTD group than in the AD group. The difference in memory and cognitive functions simply resulted in the difference between the AD and FTD groups. Further studies with participants with AD and FTD with similar MMSE scores are needed to clarify the relationship between the Ap-An ratio and decline in brain function.

Since we were able to demonstrate the difference in the alpha envelope between the NC group and the AD/FTD groups, it was suggested that the alpha envelope and the Ap-An ratio could be neurophysiological markers to detect and evaluate brain function related to E-I regulation.

## Limitations

In the present study, we considered the second-order derivative with respect to time of the alpha wave envelope fluctuation as a factor characterizing the envelope waveform and investigated whether it could be related to brain dysfunction. Although the relationship with previous mathematical models was considered, this study was approached from the clinical EEG analysis. We did not consider a direct correspondence with the NMMs or their equations in the present report, but a mathematical model to explain this phenomenon should also be considered. The correspondence with the pathophysiology of the neural activity occurring was not clear, as was the physiological implication. It would be necessary to verify inductively the characteristics and changes of the alpha envelope by evaluating it in different pathological conditions. As for findings in normal participants, this study used only participants with a limited age range over the late 40s. Senile changes may be mixed in the NC group. Aging and developmental effects, gender differences, dominant hand effects, etc. should be considered in further studies to determine the normal alpha envelope characteristic. Intra-individual variation and the presence or absence of task-related changes in the envelope should also be clarified. Because the analysis in this study was novel, better variations in analysis windows and envelope generation methods might need to be developed for better analysis in future studies. We expected that measurements in various pathological conditions would clarify the meaning of the envelope. Although many future studies are needed, we believed that a new method with envelope analysis, as found in the present study, could be potentially useful for evaluating brain activity.

## Conclusion

We conducted the present study a new method of analyzing the envelope of alpha activity of the EEG. The Ap-An ratio of the alpha activity envelope in the present study, which might related with inhibitory and excitatory functions of alpha oscillations, could be a useful biomarker to assess brain function, although this novel analysis method did not involve a high degree of complexity. As shown in the present results, the envelope amplitude analysis could provide new insights into brain activity in normal participants and participants with dementia. We believe that the envelope of alpha activity should be studied not only in healthy individuals, but also in various brain disorders, including functional disorders that did not show abnormalities in clinical EEG reading.

## Supporting information

**S1 Table. Information of all participants.**
(DOCX)

## Author Contributions

**Conceptualization:** Misako Sano, Katsuyuki Iwatsuki, Hitoshi Hirata, Minoru Hoshiyama.

**Data curation:** Misako Sano, Minoru Hoshiyama.

**Funding acquisition:** Hitoshi Hirata, Minoru Hoshiyama.

**Methodology:** Minoru Hoshiyama.

**Supervision:** Katsuyuki Iwatsuki, Hitoshi Hirata.

**Writing – original draft:** Misako Sano, Minoru Hoshiyama.

**Writing – review & editing:** Yuko Nishiura, Izumi Morikawa, Aiko Hoshino, Jun-ichi Uemura, Katsuyuki Iwatsuki, Hitoshi Hirata, Minoru Hoshiyama.

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
