## [Decision Letter · Decision Letter 0]

1 Apr 2024

PONE-D-23-35942Analysis of the alpha activity envelope in electroencephalography in relation to the ratio of excitatory to inhibitory neural activityPLOS ONE

Dear Dr. Hoshiyama,

Thank you for submitting your manuscript to PLOS ONE. After careful consideration, we feel that it has merit but does not fully meet PLOS ONE’s publication criteria as it currently stands. Therefore, we invite you to submit a revised version of the manuscript that addresses the points raised during the review process.

Two experts in the field have carefully reviewed the manuscript entitled, "Analysis of the alpha activity envelope in electroencephalography in relation to the ratio of excitatory to inhibitory neural activity". Their comments are appended below.

The first reviewer acknowledged that the manuscript is fairly well-written with leaving several serious concerns which should be considered before publication.

On the other hand, the second referee pointed out that there is the fundamental concern regarding the hypothesis the author posted.

We look forward to receiving your revised manuscript.

Kind regards,

Manabu Sakakibara, Ph.D.

Academic Editor

PLOS ONE

Journal Requirements:

"Minoru Hoshiyama reports financial support was provided by JSPS Grant-in-Aid for Scientific Research (C) (20K07881) and Japan Agency for Medical Research and Development. The authors declare that they have no known competing financial interests or personal relationships that could have appeared to influence the work reported in this paper. "

"This study was partially supported by JSPS Grant-in-Aid for Scientific Research (C) (20K07881) and AMED-CREST (23gm1510005h0003) in Japan."

"Minoru Hoshiyama reports financial support was provided by JSPS Grant-in-Aid for Scientific Research (C) (20K07881) and Japan Agency for Medical Research and Development. The authors declare that they have no known competing financial interests or personal relationships that could have appeared to influence the work reported in this paper. "

Reviewers' comments:

Reviewer's Responses to Questions

**Comments to the Author**

1. Is the manuscript technically sound, and do the data support the conclusions?

Reviewer #1: Yes

Reviewer #2: Partly

2. Has the statistical analysis been performed appropriately and rigorously? 

Reviewer #1: Yes

Reviewer #2: I Don't Know

3. Have the authors made all data underlying the findings in their manuscript fully available?

Reviewer #1: Yes

Reviewer #2: Yes

4. Is the manuscript presented in an intelligible fashion and written in standard English?

Reviewer #1: Yes

Reviewer #2: No

5. Review Comments to the Author

Reviewer #1: General Comment:

I thank the authors for the opportunity to review this work. The amount of work put into this research is evident. For this reason, I hope to offer suggestions which will increase the effectiveness of the paper.

The paper offered an interesting new technique to evaluate EEG-based alpha activity within the human brain – one in which has not been thoroughly explored before. Although the paper was communicated coherently overall, minor specifications were missing which impact the reproducibility of the work. For that reason, I have highlighted these recommended changes along with some minor comments for the author to consider, for the work to be more impactful overall.

Abstract:

The abstract covers most of the necessary components in order to summarize the research but could have been a bit more quantitative in nature. Providing the supporting quantitative results would help solidify the significance of the findings and would render the paper as more impactful overall.

Page 2 – The abstract states that alpha waves are “the major components of cortical activity in the electroencephalography (EEG)”. Although this is not completely incorrect, I believe it to be critical to state that this is in reference to waking human brain activity. For instance, it is well established that during resting states, lower frequency bands (i.e., delta and theta) dominate neural activity in the brain, therefore this distinction is necessary.

Introduction:

The introduction provided a good balance of background knowledge and rational for the research. A few minor comments on the introduction include:

Page 7 – Line beginning with “the waxing and waning” - while introducing such concepts it, would be helpful if the authors provided more details on “the summations of multiple dynamics”.

Page 10 – The introduction ended very strongly. The authors were very clear in what their aims were, what the paper entailed, and the implications of the work.

Methods:

The methodology was presented in a logical structure, although was missing a few minor details. Further details are required on the following:

Page 10 – Please state why the authors opted to use eyes closed data.

For instance, the choice of using eyes closed EEG data is a great one as it avoids the need to discriminate neuronal signals associated with mechanisms of visual processing and oculomotor artifact (such as eye-blinks and eye-movements). For this reason, it would be of great value if the authors stated this, along with any other reasons why eyes closed data were used.

Page 12 section 2.2 – Please state why only 60 seconds of EEG data were evaluated.

Page 12 sections 2.2 – Specify which key features were involved in this selection process. In other words, please state which markers of noise were evaluated.

Page 12 – data analysis sections should be denoted as section 2.3.

Page 12 sections 2.3 – Were the peak alpha frequencies evaluated on an individual basis, e.g., in the form of the Individual Alpha Peak Frequency (IAPF)? If so, please specify. If not, please indicate why IAPFs were not used.

Page 13 section 2.3 - Please state the process involved in the signals being rectified.

Additionally, it would be helpful stating the purpose/why EEG rectification was used. This would ultimately allow the research to be more well received and applied to a wider audience (i.e., by those who may not be as familiar with EEG-based research and terminology).

Page 13 section 2.3 - Which MATLAB command was used? If this is in reference to Fourier Transform, please specify.

Page 14 section 2.3 – Please clarify why “An I value was originally a negative value, but in

this study an I value was expressed as a positive value”.

Page 14 section 2.4 – A minor note, but I would suggest changing the phrase “at each electrode” to “at each electrode site” as the former refers to a single electrode whereas the latter refers to all electrodes at a single scalp location (e.g., electrode site F3, etc.).

Discussion:

I commend the authors on the work, as the findings seem novel and were fascinating to learn about. In order to more accurately represent the value of the findings, I recommend the following minor alterations of the discussion section:

Page 18 – I recommend putting all limitations and future work in a section toward the end of the discussion, as this would allow the findings to be presented in a stronger and more convincing manner.

Page 19 section 4.1 – When discussing the findings, I suggest tying them in with previous work which may be similar in concept (even if not of similar methodology, but of similar underlying neuronal mechanism). Doing so would strengthen this section, and thereby further substantiating the findings of this study.

Page 19 section 4.1 – Regarding the sentence: “That is, the excitatory force or second derivative that increased the amplitude was, on average, greater than the force that decreased the amplitude.” - What does this indicate on a neural level?

Page 20 section 4.1 – “Since both E and I were positively correlated with amplitude, a region of higher mean envelope amplitude, resulting from higher amplitude alpha activity, had more inhibitory than excitatory power operating.” – What are the implications of this finding, more specifically, in relation to healthy participants?

Page 21 section 4.1 – “the present study, we might observe that excitatory and inhibitory components or factors in one frequency band, i.e., the alpha frequency band, resulted from the effects of other frequency bands” – Well noted and a strong end to this section.

Page 22 section 4.2 – “We thought that this alpha envelope analysis method allowed us to evaluate the E-I ratio from a temporal sequence of the alpha rhythm obtained at a region or electrode.” – I suggest adding a point or two to this sentence as to why the authors believed this, as a means of strengthening the statement.

Page 22 section 4.2 – “less in the number of electrodes” – are the authors referring to less in all of the electrodes or specific electrodes. If the latter, please specify which electrodes this is in reference to.

Page 22 section 4.2 – “this result could simply related with the severity of dementia” – it would strengthen this statement if a previous research was used to substantiate this assumption. Are there any underlying mechanisms which could further support this statement?

Page 24 section 4.2 – “Further studies with participants with AD and FTD with similar MMSE scores are needed to clarify the relationship between the E-I ratio and deterioration of brain

function.” – Great concluding sentence.

Conclusion:

The conclusion was clear and concise, and very effective overall. With that said, I recommend moving the limitations section to before the conclusion section as I believe this would increase the effectiveness of the work by ending on a positive note.

Page 24 – the phrase "quite simple" downplays the method, making it seem less strong overall. I would suggest rephrasing this to something along the lines of "Although, this novel analysis method does not involve high degrees of complexity..", as the value of the work is presented in a more convincing and persuasive manner.

Figures:

Figure 1 – “Methods of analysis of electroencephalography (EEG) signals.” I would recommend changing this to “ A representation of the method of analysis” as this figure supports the methodology, but is not the “method” per se.

I appreciate this opportunity to review this work and hope that the constructive feedback will be of benefit to the Authors.

Reviewer #2: In this study, the authors investigate the link between the phenomenon of waxing and waning amplitude in the alpha frequency band (8-12 Hz) and the E/I (excitation/inhibition) ratio. This research goal is pursued by first extracting the envelope of the 8-12 Hz band-pass filtered signal. Consequently, the second derivative of the envelope was extracted, assuming that negative values corresponded to a higher influence of inhibitory populations, while positive values to be driven by excitatory ones. As a practical example: in this study, a higher E/I ratio is equivalent to a steeper increase of the amplitude in the alpha frequency range. The differences with respect to these parameters, and its relevance to clinical populations of patients with AD and FTD is then used as a benchmark for these EEG quantitative metrics.

Although the subject of this study is currently of great interest, I would recommend that the authors address a few major points before publication. After a brief summary of my suggestions and comments, I will detail every comment, referring to the exact manuscript line (where possible).

Authors assume that the waxing and waning of alpha frequency amplitude is driven by a dynamic disequilibrium in the balance between excitatory and inhibitory populations, irrespective of which cortical/subcortical region we are subjecting to investigation; they make this assumption by advancing that ERP studies support this view. However, it is my personal opinion that this claim needs stronger evidence. From my point of view, providing more literature that backs up this claim is at least necessary to improve the quality of the manuscript. In an ideal case, authors might want to establish their claims by providing evidence in a computational model: as they don't make any region-specific claim, I think that a neural mass model should be sufficient in this case. Without these precautions, I believe that the authors' claims are not really supported by the data presented, which also makes it very difficult to intepret their clinical results even for an expert reader.

In the manuscript, there is also some missing information that I think are very important for a thorough understanding of their work, but I'm confident that authors can fill in the gaps very quickly.

Major issue: at the current state, it's really hard to confirm that the authors' assumptions and results are backed up by the data presented. Because of this, I would suggest a few steps before publication:

- provide, at least in the introduction, literature supporting the validity of using the second derivative of the alpha frequency envelope as a proxy measure to the E/I ratio;

- ideally, showing this link in a computational model (I would suggest using a neural mass model) is a major key point is solidifying this work's results.

- addressing the minor issues, which will be detailed in the next paragraph, citing the exact manuscript page where possible.

Introduction

- page 8: "In the case of evoked

potentials, the increase or decrease in amplitude of the components was

considered to be the sum of excitation and inhibition processes that occurred

prior to the generation of the wave. Such excitation and inhibition could be

considered not only for afferent neural signals evoked by external stimuli,

but also for intrinsic neural activity, e.g., neural activity producing alpha

waves, at the cortical and subcortical levels." Reference is needed here, even if the study was already cited in the previous paragraphs.

- page 10: "We believe that this study was conducted using a new method

with potential future applications." claims of novelty are not really necessary in the introduction. However, this is my personal view and I haven't found anything against it in the journal's editorial guidelines, which makes it not really necessary to address.

Methods

- page 11. How many ICs were rejected, on average? and is there a reason for not using a CAR rereferencing before ICA?

- page 12. I'm not really sure of why a 60s window was selected for all EEG recordings. It's my understanding that the reason is probably related to too many artifacts present in these datasets, but can you elaborate more on this?

- page 13: if I'm not mistaken, authors use "differentiated" to indicate derivatives. I might be wrong, and I apologize if that's the case, but it would be better to mantain a certain degree of consistency when using technical jargon.

- page 14. Some important details are missing in the description of the statistical methods used here. In order: which statistical test did you use to compare the metrics presented here? which type of correlation did you apply (Pearson, Spearman's, etc...)? how was significance for the correlation computed?

Results

- page 15. Maybe MMSE (mini mental state exam) needs to be introduced before the results.

Discussion

- page 19, "The mean E-I

ratio was 1.38 for normal participants using the present analysis". Is this value comparable to what has been found in other studies, even if they used different methods?

- page 19, "These results suggested that the waxing and waning of the alpha envelope was regulated by a pattern of balance between excitatory and inhibitory forces." In light of my earlier comments, I would suggest to advance these claims only if the link between E/I ration and alpha amplitude are clarified.

- page 20 I'm not sure of what "balistic" means, in this context. Please clarify.

- page 21, " In the present study, we might observe that

excitatory and inhibitory components or factors in one frequency band, i.e.,

the alpha frequency band, resulted from the effects of other frequency bands." I'm not really sure of this sentence's meaning. It would be very kind of the authors if this could be rephrased to improve readabilty.

Figures

Please write titles for each part of the figure, as for instance in Figure 2's graphs. As of now, it's really hard to follow what's depicted in the figures.

6. PLOS authors have the option to publish the peer review history of their article (what does this mean?). If published, this will include your full peer review and any attached files.

Reviewer #1: No

Reviewer #2: No

---

## [Author Response · Author response to Decision Letter 0]

25 Apr 2024

Responses to the reviewers

First I would like thank you very much for your thoughtful review. We have revised the manuscript extensively, as far as we could, according to the reviewer’s suggestions. Revised parts of the manuscript were typed in red. 

Major revisions are:

1) Introduction has been fully revised. We have described the background of the analysis method used in this study, the mathematical model involved, and why we chose this analysis method. All descriptions of poorly related evoked brain responses have been removed.

2) The term "E-I ratio" used in the previous version has been changed to "Ap-An ratio" to avoid confusion of the readers, since the previous E and I values were not identical with the excitation and inhibition functions of alpha waves, but only values for second derivatives of the alpha envelope in time.

3) The Discussion sections have also been extensively revised by adding sentences about previous related studies and studies of neural models.

Reviewer #1: General Comment:

I thank the authors for the opportunity to review this work. The amount of work put into this research is evident. For this reason, I hope to offer suggestions which will increase the effectiveness of the paper.

The paper offered an interesting new technique to evaluate EEG-based alpha activity within the human brain – one in which has not been thoroughly explored before. Although the paper was communicated coherently overall, minor specifications were missing which impact the reproducibility of the work. For that reason, I have highlighted these recommended changes along with some minor comments for the author to consider, for the work to be more impactful overall.

Abstract:

The abstract covers most of the necessary components in order to summarize the research but could have been a bit more quantitative in nature. Providing the supporting quantitative results would help solidify the significance of the findings and would render the paper as more impactful overall.

Response: We have revised the abstract to include quantitative results. Values with significant difference have been added in the abstract.

Page 2 – The abstract states that alpha waves are “the major components of cortical activity in the electroencephalography (EEG)”. Although this is not completely incorrect, I believe it to be critical to state that this is in reference to waking human brain activity. For instance, it is well established that during resting states, lower frequency bands (i.e., delta and theta) dominate neural activity in the brain, therefore this distinction is necessary.

Reply: We agreed that your points were important. We revised the sentences. The first sentence of the Introduction has also been revised.

(Abstract)

Alpha waves, one of the major components of resting and awake cortical activity in human electroencephalography (EEG), are known to show waxing and waning, but this phenomenon has rarely been analyzed.

(Introduction)

The most characteristic activity of the cerebral cortex observed in human electroencephalography (EEG) at rest and during wakefulness are alpha waves [1,2].

Introduction:

The introduction provided a good balance of background knowledge and rational for the research. A few minor comments on the introduction include:

Page 7 – Line beginning with “the waxing and waning” - while introducing such concepts it, would be helpful if the authors provided more details on “the summations of multiple dynamics”.

Response: We have added sentences in Introduction by adding a reference.

From the perspective of brain structure, alpha wave fluctuations, oscillations, have been thought to occur as cortical rhythms, or as part of a neural circuit that forms between cortices, or outside the cortex, such as in the thalamus [3-9]. Apart from the mechanism of generation, alpha wave oscillations have been variously reported to be associated with cognitive mechanisms and diseases [10-12].

Page 10 – The introduction ended very strongly. The authors were very clear in what their aims were, what the paper entailed, and the implications of the work.

Thank you for your comment. Most of Introduction has been revised, but the last sentence has been retained.

Methods:

The methodology was presented in a logical structure, although was missing a few minor details. Further details are required on the following:

Page 10 – Please state why the authors opted to use eyes closed data.

For instance, the choice of using eyes closed EEG data is a great one as it avoids the need to discriminate neuronal signals associated with mechanisms of visual processing and oculomotor artifact (such as eye-blinks and eye-movements). For this reason, it would be of great value if the authors stated this, along with any other reasons why eyes closed data were used.

Response: Thank you for your thoughtful comments. We have added sentences to explain why we chose the closed eye data at the beginning of the section.

We chose to use this open dataset because it provided stable EEG recordings during resting, awake, and closed-eye conditions in the sitting position with sufficient recording time, more than 10 minutes for each participant [33-34]. Closed-eye recording was advantageous not only to avoid blink artifacts and brain activity associated with visual stimuli and eye movements, but also to maintain a constant resting state, especially in patients with dementia. The seated position was valuable in preventing participants from falling asleep.

Page 12 section 2.2 – Please state why only 60 seconds of EEG data were evaluated.

Response: Thank you for your important comment. We have added an explanation to the section 2.3., although it is a bit lengthy. We have also added sentences about the methods in the Limitations section. Since the present analysis was novel, better variants in terms of analysis methods, including analysis window and envelope creation methods, may need to be developed in further studies.

(In the section 2.3. Selection of epochs for the present analysis)

It has been reported that the envelope of alpha waves contained fluctuations with periods ranging from a few seconds to tens of seconds [14]. This study was not a frequency analysis of the envelope, but an analysis of the increase or decrease in amplitude produced by the peaks of alpha waves. As shown in Fig 1, a few peaks of the alpha envelope were obtained in 1 second, and more than several dozen peaks of the envelope were included in a 60-second period, which was considered sufficient for the present analysis. A practical reason was that even in the 10-min resting EEG recordings of each participant, there were only one or two 60-second epochs with low artifacts such as motion. Not all participants had multiple suitable continuous 60-second recordings for analysis, and we had to select a 60-second epoch from the 10-min recording, particularly for participants with dementia (S1 Table, Supporting Information).

Page 12 sections 2.2 – Specify which key features were involved in this selection process. In other words, please state which markers of noise were evaluated.

Response: We have added sentences in the section 2.3.

Although preprocessing and artifact rejection algorithms for electrocardiogram and eye blink artifacts were applied to the datasets, some electrical artifacts were present, as mentioned by Miltiadous et al. [33-35], possibly due to extra-brain electrical noise or some motion artifact. Therefore, we manually selected a 60-second window without significant noise for each participant. During the manual selection of the 60-second epoch, we selected the 60-second epochs based on the following criteria: 1) electromyography (EMG) burst, 2) square-wave noise, 3) abrupt baseline shift, and 4) noise greater than 300 µV. Even if the amplitude was less than 300 µV, the waves that were considered electrical artifacts because of their shape were omitted from the epochs. The epochs selected for each participant were listed in S1 Table in the Supporting Information.

Page 12 – data analysis sections should be denoted as section 2.3.

Response: Section numbers have been removed, according to the journal format.

Page 12 sections 2.3 – Were the peak alpha frequencies evaluated on an individual basis, e.g., in the form of the Individual Alpha Peak Frequency (IAPF)? If so, please specify. If not, please indicate why IAPFs were not used.

Response: We measured the peak alpha frequency at each electrode in each participant. We have revised the sentence in the section 2.4. Although the mean peak frequency at each electrode is shown in Table 1, we have added the value of the peak alpha frequency at the occipital area, at O1 and O2 electrodes, in the table in the Supporting Information.

The peak frequency in the alpha frequency band in the power spectrum density was calculated at each electrode in each participant using Welch's method, which divided the signals into 50% overlapping windows for each second.

The mean amplitude expressed as root mean square (RMS) of alpha activity, the peak frequency of alpha activity for 60 seconds analyzed are shown in Table 1. The peak alpha frequency in the occipital areas, at each electrode O1 and O2 where the alpha waves were typically found [47], for each participant was shown in S1 Table in the Supporting Information.

Page 13 section 2.3 - Please state the process involved in the signals being rectified. Additionally, it would be helpful stating the purpose/why EEG rectification was used. This would ultimately allow the research to be more well received and applied to a wider audience (i.e., by those who may not be as familiar with EEG-based research and terminology).

Response: We have added sentences for explanation of the process and reason.

The baseline was offset by the signal level averaged over 60 seconds, and the signal was rectified by taking the absolute value of the sampling points. In the present study, we analyzed the amplitude of the envelope formed by the alpha wave peaks. The envelope could be plotted for positive and negative alpha wave peaks relative to the baseline. Since alpha waves have been observed as oscillations on scalp EEG [46], the positive and negative deviations of alpha waves were treated as caused by similar temporal sequences of oscillations in the present study. To obtain more temporal information about the number of peaks per unit time, the waveforms were rectified to create the envelopes.

Page 13 section 2.3 - Which MATLAB command was used? If this is in reference to Fourier Transform, please specify.

Response: Recent MATLAB have provided command for envelope, and we used one of the commands for envelope calculation.

[yupper, ylower] = envelope (x, n, ‘peak’)

envelope(x) returns the upper and lower envelopes of the input sequence x as the magnitude of its analytic signal. The envelope(x, n, 'peak') command returns the peak envelopes of the sequence signal. The envelopes are determined using spline interpolation over local maxima separated by at least n samples. The analytic signal of x is found using the discrete Fourier transform as implemented in the "hilbert" command. The function first removes the mean of x and then adds it back after calculating the envelopes.

Since we calculated the Ap and An values at each sample point, we did not average the signal, i.e. n=1 in the command, to create the envelope.

These sentences can be found in the MATLAB tutorial, we have added a sentence about the MATLAB command we used.

The MATLAB "envelope (‘peak’)" command to create an envelope included the Hilbert transform processes.

Page 14 section 2.3 – Please clarify why “An I value was originally a negative value, but in this study an I value was expressed as a positive value”.

Response: Since the negative acceleration values were defined as An values, the original An values were all negative. Since the Ap-An ratio is a ratio of the magnitude of excitation and inhibition, a positive or negative sign is not meaningful. The description of the positive and negative signs is misleading, so the description has been changed to clarify the values.

The envelope of the alpha wave time course was second differentiated at each sampling point. We defined an absolute value of a positive and a negative value of the second differentiation at each sampling point as acceleration positive (Ap) and negative (An) values, respectively, and the Ap-An ratio was a ratio of Ap to An values at each sampling point. 

Ap value = | positive second differentiated value | (3)

An value = | negative second differentiated value | (4)

Ap-An ratio = [ Ap value ] / [ An value ] (5)

The average of the Ap and An values and the Ap-An ratio for 60 seconds were obtained at each electrode. The root mean square (RMS) of the alpha activity for 60 seconds at each electrode was calculated as the envelope amplitude.

Page 14 section 2.4 – A minor note, but I would suggest changing the phrase “at each electrode” to “at each electrode site” as the former refers to a single electrode whereas the latter refers to all electrodes at a single scalp location (e.g., electrode site F3, etc.).

Response: the phrase has been changed as you could suggest.

The envelope amplitude, the Ap-An ratio, and Ap and An values were compared between groups, AD and NC, and FTD and NC groups at each electrode site.

Pearson's correlation coefficient between envelope amplitude and Ap and An values and Ap-An ratio was compared between groups at each electrode site.

Discussion:

I commend the authors on the work, as the findings seem novel and were fascinating to learn about. In order to more accurately represent the value of the findings, I recommend the following minor alterations of the discussion section:

Page 18 – I recommend putting all limitations and future work in a section toward the end of the discussion, as this would allow the findings to be presented in a stronger and more convincing manner.

Response: We have moved the sentences to the Limitation section at the end of the manuscript. The sentences in the Limitation section have been extensively revised.

Page 19 section 4.1 – When discussing the findings, I suggest tying them in with previous work which may be similar in concept (even if not of similar methodology, but of similar underlying neuronal mechanism). Doing so would strengthen this section, and thereby further substantiating the findings of this study.

Thank you very much for your important comment. We have completely revised the introduction and provided methodological background. The discussion of the envelope of alpha activity and the Ap-An ratio in healthy participants has also been extensively revised to show previous studies with related concepts.

Page 19 section 4.1 – Regarding the sentence: “That is, the excitatory force or second derivative that increased the amplitude was, on average, greater than the force that decreased the amplitude.” - What does this indicate on a neural level?

This sentence has been deleted. The reason was that the average of the magnitude of the acceleration does not necessarily equal the sum of the applied forces.

Page 20 section 4.1 – “Since both E and I were positively correlated with amplitude, a region of higher mean envelope amplitude, resulting from higher amplitude alpha activity, had more inhibitory than excitatory power operating.” – What are the implications of this finding, more specifically, in relation to healthy participants?

This sentence was also deleted because its meaning was ambiguous. The discussion on healthy people has been substantially revised.

Page 21 section 4.1 – “the present study, we might observe that excitatory and inhibitory components or factors in one frequency band, i.e., the alpha frequency band, resulted from the effects of other frequency bands” – Well noted and a strong end to this section.

Thank you for your comment, but we have revised the sentences according to the other reviewer’s comment.

In the present study, only alpha wave fluctuations were analyzed, so the relationship with other frequencies was not considered. It should be noted that the alpha wave fluctuations were not only controlled by 

---

## [Decision Letter · Decision Letter 1]

10 May 2024

PONE-D-23-35942R1Analysis of the alpha activity envelope in electroencephalography in relation to the ratio of excitatory to inhibitory neural activityPLOS ONE

Dear Dr. Hoshiyama,

Thank you for submitting your manuscript to PLOS ONE. After careful consideration, we feel that it has merit but does not fully meet PLOS ONE’s publication criteria as it currently stands. Therefore, we invite you to submit a revised version of the manuscript that addresses the points raised during the review process.

The original referees have carefully reviewed the revision. They are almost satisfied with the manuscript with leaving some minor concerns which should be considered before publication.

I will make the final decision after receiving the necessary revised manuscript and the reply mail to each critique.

We look forward to receiving your revised manuscript.

Kind regards,

Manabu Sakakibara, Ph.D.

Academic Editor

PLOS ONE

Journal Requirements:

Reviewers' comments:

Reviewer's Responses to Questions

**Comments to the Author**

1. If the authors have adequately addressed your comments raised in a previous round of review and you feel that this manuscript is now acceptable for publication, you may indicate that here to bypass the “Comments to the Author” section, enter your conflict of interest statement in the “Confidential to Editor” section, and submit your "Accept" recommendation.

Reviewer #1: All comments have been addressed

Reviewer #2: All comments have been addressed

2. Is the manuscript technically sound, and do the data support the conclusions?

Reviewer #1: Yes

Reviewer #2: Yes

3. Has the statistical analysis been performed appropriately and rigorously? 

Reviewer #1: Yes

Reviewer #2: Yes

4. Have the authors made all data underlying the findings in their manuscript fully available?

Reviewer #1: (No Response)

Reviewer #2: Yes

5. Is the manuscript presented in an intelligible fashion and written in standard English?

Reviewer #1: Yes

Reviewer #2: Yes

6. Review Comments to the Author

Reviewer #1: I first would like to commend the authors for the improvement in the manuscript. With that said, I believe there to be a view minor points to address before being ready for acceptance. These are the following:

Abstract:

Line 34-39: I strongly recommend breaking the sentence up into two to increase readability. E.g., Second sentence starting line 37 "This was done in 36....."

Highlights:

Line 71: I recommend changing the starting of the second point so it varies slightly from the first (i.e., not starting with "a new" once again.

Introduction:

Line 91-98: Very well improved!

Line 105-108: Once again, a bit difficult to read. I recommend re-structuring/ splitting into two sentences.

Line 114: Move this to the sentence after, e.g., by stating "Such aims were based upon previous work, which defines.... (then state where it is based upon).

Line 136: I would move this to the discussion/limitation section as keeping it in the introduction weakens the argument of the current exploration.

Line 167: This sentence comes off weak. Perhaps saying "This study is believed...." rather than "thought". A minor recommendation but wording does strengthen the argument of the work to the reader!

Methods:

The authors have put in evident work to drastically improve the methods section, making it a lot more cohesive and sound. With that stated, I highly recommend adding some visual support (e.g., figures or diagrams) which support the research/analysis. As the concepts explore are fairly complex in nature, I strongly believe a visual component will strengthen receptivity of the work immensely.

I hope that these minor suggestions help improve the manuscript further beyond the already evident improvement!

Reviewer #2: All comments were adequately addressed by the authors.

I would like to thank the authors for the patience taken to revise their work. I also suggest, as a future direction, to investigate the mechanisms underlying the acceleration/deceleration of the envelope in the alpha band, as it might provide an interesting window into neural activity.

7. PLOS authors have the option to publish the peer review history of their article (what does this mean?). If published, this will include your full peer review and any attached files.

Reviewer #1: No

Reviewer #2: No

---

## [Author Response · Author response to Decision Letter 1]

22 May 2024

Responses to the reviewers

Thank you very much for your very thoughtful peer review.

We have again revised the manuscript carefully according to the reviewer’s comments. In particular, we have added a diagram illustrating the concept of this study.

Reviewer #1: I first would like to commend the authors for the improvement in the manuscript. With that said, I believe there to be view minor points to address before being ready for acceptance. These are the following:

Abstract:

Line 34-39: I strongly recommend breaking the sentence up into two to increase readability. E.g., Second sentence starting line 37 "This was done in 36....."

The sentence has been revised.

The alpha wave envelope was subjected to secondary differentiation. This gave the positive (acceleration positive, Ap) and negative (acceleration negative, An) values of acceleration and their ratio (Ap-An ratio) at each sampling point of the envelope signals for 60 seconds. This analysis was performed on 36 participants with Alzheimer's disease (AD), 23 with frontotemporal dementia (FTD) and 29 age-matched healthy participants (NC) whose data were provided as open datasets.

Highlights:

Line 71: I recommend changing the starting of the second point so it varies slightly from the first (i.e., not starting with "a new" once again.

The highlight #2 has been changed.

#2: From the perspective of excitatory and inhibitory regulation, a novel aspect of cortical alpha oscillations was proposed

Introduction:

Line 91-98: Very well improved!

We would like to thank the reviewers for their help.

Line 105-108: Once again, a bit difficult to read. I recommend re-structuring/ splitting into two sentences.

The sentence has been revised and split into two sentences.

One hypothesis concerning the timing of the inhibitory and excitatory effects of alpha oscillations, including the waxing and waning phenomenon, proposed that the rhythmicity of alpha waves mediates inhibition [19-21,23]. Another suggested that waxing and waning resulted from the summation of multiple periodic phenomena [14].

Line 114: Move this to the sentence after, e.g., by stating "Such aims were based upon previous work, which defines.... (then state where it is based upon).

The sentences have been revised.

The aim of the present study was to show characteristics of alpha wave fluctuation at rest, i.e. 1) the ratio of increasing and decreasing forces that form the alpha oscillation and its relationship with amplitude, and 2) the difference of these values between healthy participants and patients with dementia in whom cognitive dysfunction occurred. These aims were based on previous work suggesting that the waxing and waning of alpha waves is regulated in an amplitude-dependent manner [24, 25] and that this functional regulation may be disrupted in patients with dementia [36-39].

Line 136: I would move this to the discussion/limitation section as keeping it in the introduction weakens the argument of the current exploration.

We have moved part of the sentence into the section of limitation.

(In Limitations)

We did not consider a direct correspondence with the NMMs or their equations in the present report, but a mathematical model to explain this phenomenon should also be considered.

Line 167: This sentence comes off weak. Perhaps saying "This study is believed...." rather than "thought". A minor recommendation but wording does strengthen the argument of the work to the reader!

Thank you for your suggestion. The sentence has been revised.

This study was believed that the present analysis would also allow us to detect the destruction of the function of alpha activity in various disorders.

Methods:

The authors have put in evident work to drastically improve the methods section, making it a lot more cohesive and sound. With that stated, I highly recommend adding some visual support (e.g., figures or diagrams) which support the research/analysis. As the concepts explore are fairly complex in nature, I strongly believe a visual component will strengthen receptivity of the work immensely.

We have added a figure which illustrated an overview the method of the present study.

I hope that these minor suggestions help improve the manuscript further beyond the already evident improvement!

Thank you again for your detailed peer review.

Reviewer #2: All comments were adequately addressed by the authors.

I would like to thank the authors for the patience taken to revise their work. I also suggest, as a future direction, to investigate the mechanisms underlying the acceleration/deceleration of the envelope in the alpha band, as it might provide an interesting window into neural activity.

Thank you for your kind comments. Thanks to your suggestions, we have been able to improve the manuscript. We will proceed to investigate the underlying mechanisms of this phenomenon and develop a mathematical model.

---

## [Editor Report · Decision Letter 2]

24 May 2024

Analysis of the alpha activity envelope in electroencephalography in relation to the ratio of excitatory to inhibitory neural activity

PONE-D-23-35942R2

Dear Dr. Hoshiyama,

We’re pleased to inform you that your manuscript has been judged scientifically suitable for publication and will be formally accepted for publication once it meets all outstanding technical requirements.

Kind regards,

Manabu Sakakibara, Ph.D.

Academic Editor

PLOS ONE
---

## [Editor Report · Acceptance letter]

3 Jun 2024

PONE-D-23-35942R2 

PLOS ONE

Dear Dr. Hoshiyama, 

I'm pleased to inform you that your manuscript has been deemed suitable for publication in PLOS ONE. Congratulations! Your manuscript is now being handed over to our production team.

Kind regards, 

on behalf of

Dr. Manabu Sakakibara 

Academic Editor

PLOS ONE